# Integrative inference of subclonal tumour evolution from single-cell and bulk sequencing data

Salem Malikic[1,2,6], Katharina Jahn[3,4,6], Jack Kuipers[3,4,6], S. Cenk Sahinalp[5,7] & Niko Beerenwinkel [3,4,7]

Understanding the clonal architecture and evolutionary history of a tumour poses one of the key challenges to overcome treatment failure due to resistant cell populations. Previously, studies on subclonal tumour evolution have been primarily based on bulk sequencing and in some recent cases on single-cell sequencing data. Either data type alone has shortcomings with regard to this task, but methods integrating both data types have been lacking. Here, we present B-SCITE, the first computational approach that infers tumour phylogenies from combined single-cell and bulk sequencing data. Using a comprehensive set of simulated data, we show that B-SCITE systematically outperforms existing methods with respect to tree reconstruction accuracy and subclone identification. B-SCITE provides high-fidelity reconstructions even with a modest number of single cells and in cases where bulk allele frequencies are affected by copy number changes. On real tumour data, B-SCITE generated mutation histories show high concordance with expert generated trees.

[1] School of Computing Science, Simon Fraser University, Burnaby V5A 1S6 BC, Canada. [2] Vancouver Prostate Centre, Vancouver V6H 3Z6 BC, Canada. [3] Department of Biosystems Science and Engineering, ETH Zurich, Basel 4058, Switzerland. [4] Swiss Institute of Bioinformatics, Lausanne 1015, Switzerland. [5] Department of Computer Science, Indiana University, Bloomington 47405 IN, USA. [6]These authors contributed equally: Salem Malikic, Katharina Jahn, Jack Kuipers. [7]These authors jointly supervised this work: S. Cenk Sahinalp, Niko Beerenwinkel. Correspondence and requests for materials should be addressed to S.C.S. (email: cenksahi@indiana.edu) or to N.B. (email: niko.beerenwinkel@bsse.ethz.ch)

Cancer is a genetic disease that develops through a branched evolutionary process[1]. It is characterised by the emergence of genetically distinct subclones through the random acquisition of mutations at the level of single cells and shifting prevalences at the subclone level through selective advantages purveyed by driver mutations. This interplay creates complex mixtures of tumour-cell populations, which exhibit different susceptibilities to targeted cancer therapies and are suspected to be the cause of treatment failure[2,3]. Therefore, it is of great interest to obtain a better understanding of the evolutionary histories of individual tumours and their subclonal composition[4].

Most genetic analyses of tumours are currently based on next-generation sequencing data of bulk tumour samples. Such data provide indirect measurements of the subclonal tumour composition in the form of aggregate total and variant read counts measured across hundreds of thousands or millions of cells. A large number of approaches have been published over the last few years that try to identify subclones, their frequencies and in some cases, their phylogenetic relationships by deconvolving

these aggregate data[5–23]. However, the underlying statistical problem is underdetermined[24,25], and single-nucleotide variants (SNVs) with similar variant allele frequencies (VAFs) are automatically clustered into a single subclone. This inevitably leads to incorrect phylogenies for tumours with multiple distinct subclones of similar prevalences as illustrated in Fig. 1. The aggregate sequencing data additionally pose a limitation to the achievable tree resolution, as mutational signals of smaller subclones cannot be distinguished from noise[26] and therefore not be reliably represented in the tree. Sequencing multiple samples from the same tumour and increasing the coverage can to some extent mitigate these issues, but is not always practicable.

Another solution is the use of single-cell sequencing (SCS) data which provide mutation profiles of individual cells, such that the phylogeny can be directly inferred without any form of deconvolution. The main challenge here instead are the high levels of noise found in SCS data that are primarily introduced during DNA amplification, a necessary step to obtain sufficient DNA material for sequencing. False negatives are the most prevalent

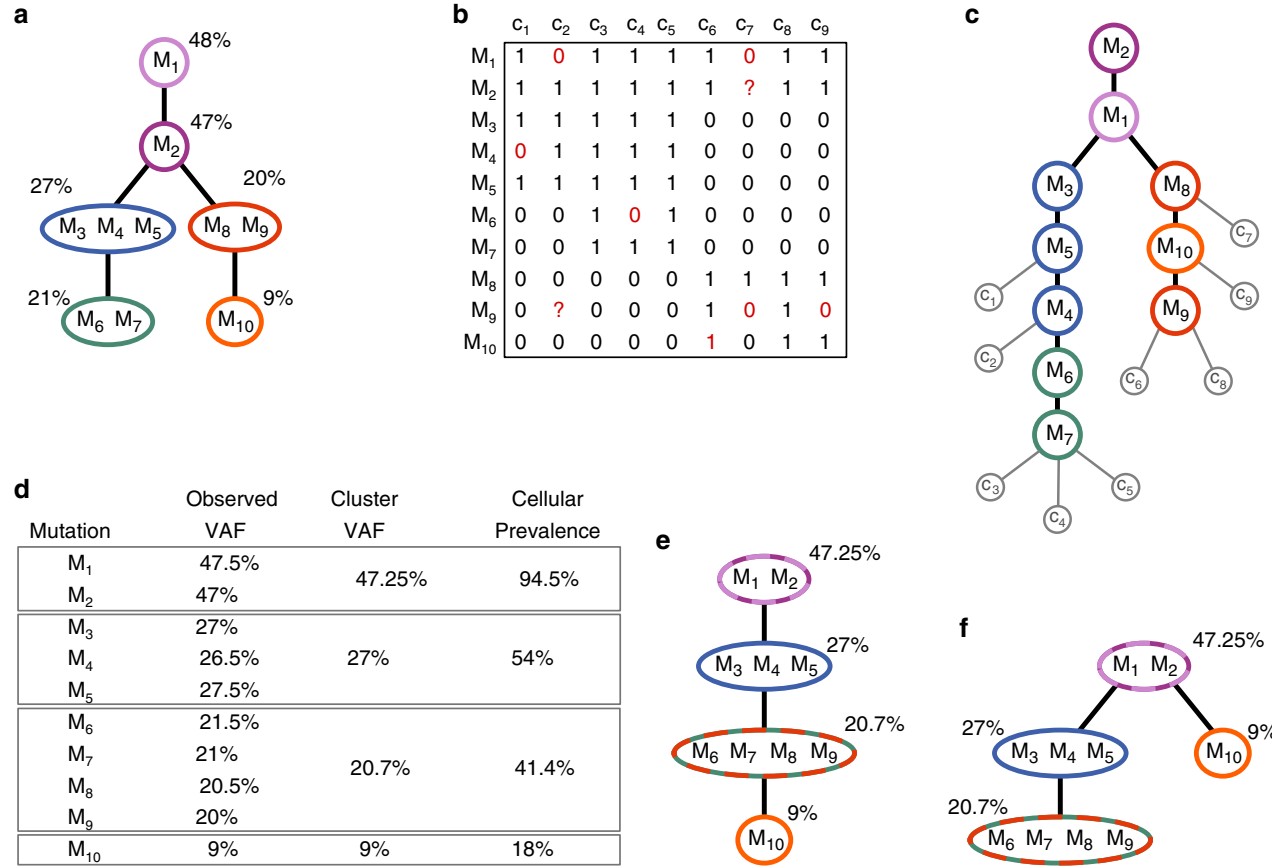

**Fig. 1** Comparison of inferred mutation histories based on single-cell and bulk-sequencing data. **a** Ground truth clonal tree with mutations $M_1, ..., M_{10}$: the coloured nodes represent the subclones, and the tree structure indicates the partial temporal order in which the subclones emerged (from top to bottom). Each subclone contains the mutations it acquired in comparison with its parent and is annotated with the mean VAF of these mutations. (For a heterozygous mutation in a copy-number-neutral region, the VAF is half the mutation's cellular prevalence.) **b** Cell mutation profiles obtained from the SCS data for nine cells $c_1, ..., c_9$: '1' indicates the observed presence of a mutation, and '0' absence. A '?' indicates a missing data point (NA), e.g., due to insufficient coverage. The red '0's are false negatives (e.g., drop-out events), the red '1' indicates a false positive. Due to these errors, the mutation matrix defines no perfect phylogeny. **c** Inferred single-cell mutation tree annotated with single-cell placements. Not all cells can be placed, such that their observed mutation profile matches with the mutations acquired along the lineage from the root to their attachment point. The branching point of the ground truth tree is inferred correctly, due to the strong signal that the red/orange and blue/green mutations do not occur in the same cell. However, mutation order in linear segments is not reliably inferred from the SCS data; especially in the right branch, a mutation with low prevalence ($M_{10}$) is placed above a more prevalent mutation ($M_9$) due to errors in the mutation profiles of cells $c_6$, $c_7$ and $c_9$. **d** Variant allele frequencies obtained from bulk sequencing. VAF-based clustering of mutations leads to merging of subclones. **e**, **f** Both clonal trees are compatible with the VAFs and the clustering inferred in (**d**). Due to the clustering and incompatible VAFs, the correct branching between the blue and red subclone is not inferred, but in both trees, mutation ordering is consistent with their true prevalences

error type due to allelic dropout, but also false positives occur when an error is introduced early in the amplification. Further noise stems from doublet-mutation profiles, which occur when two cells are accidentally sequenced together[27]. Classic approaches for phylogeny reconstruction are not suitable for dealing with these SCS-specific noise profiles, and a number of probabilistic approaches have been developed to specifically account for the error types found in SCS data[28–32].

A major difference between the evolutionary histories of tumours inferred from bulk and SCS data is that the former typically are clonal trees where mutations with similar frequencies are clustered together and represented in a single tree node (Fig. 1e, f), while trees derived from SCS data are fully resolved trees that can be either cell lineage trees, binary trees where the cells form the leaves and mutations occur along tree branches, or mutation trees (Fig. 1c) that depict the partial temporal order in which mutations were acquired[33]. For cell lineage trees, a heuristic has been proposed for clustering cells into clones in a post-processing step[29], which results in trees that are closer to bulk clonal trees.

Another difference, as illustrated in Fig. 1, is that the VAFs obtained from a bulk sample are well suited for inferring the temporal order of mutations (by ordering mutations with respect to decreasing VAF), but of limited use for the identification of branching events. On the contrary, single-cell genotypes obtained from SCS data have a lot of strength to infer branching events, but due to unobserved ancestral states and high noise levels, may give ambiguous or no signals on the temporal order of mutations in linear segments of the inferred mutation trees.

As the strengths and weaknesses of single-cell and bulk-sequencing data are to a large extent complementary with respect to phylogeny inference, using both data types for a joint inference should improve our understanding of subclonal tumour evolution over using each type of data alone. It has already been shown that clustering of mutations into subclones from bulk-sequencing data can be informed by single-cell genotypes to obtain more accurate results[34]. In this work, we present B-SCITE, a probabilistic approach for the inference of tumour phylogenies from combined single-cell and bulk-sequencing data. We show in a comprehensive simulation study that B-SCITE systematically outperforms competing approaches in terms of accurate tree reconstruction and mutation clustering. It performs particularly well in difficult cases, where only a small number of single cells is available, or when bulk read counts are affected by copy-number changes. We also show that for real tumour data, B-SCITE provides mutation histories that are in better congruence with expert-generated trees than tools relying only on a single data type.

## Results

**Method overview**. We developed B-SCITE, a probabilistic approach for the inference of tumour mutation histories by the use of SNV data obtained from single-cell and bulk DNA sequencing (Fig. 2). Full technical details are given in the 'Methods' section. B-SCITE consists of a Markov chain Monte Carlo-based search scheme to traverse the space of possible tree topologies and a joint likelihood model for the evaluation of candidate trees based on their joint fit to the single-cell and bulk data. The single-cell data are given in the form of a mutation matrix, where each row represents a mutation, for which each column represents that mutation's state in the corresponding single cell ('0' normal state, '1' altered state and 'NA' missing data); on the other hand, the bulk data consist of the variant and total read counts of the mutated loci in each of the sequenced bulk samples. B-SCITE reports a single maximum likelihood mutation tree that can be condensed into a clonal tree by clustering linear tree segments based on VAF similarity.

**Performance assessment on simulated data**. We tested B-SCITE on a comprehensive set of simulated data. We focused initially on comparing with ddClone[34], which is, to the best of our knowledge, the only existing method that performs analysis of intra-tumour heterogeneity by jointly using bulk and SCS data. ddClone was shown to significantly outperform methods only utilising the bulk-sequencing data. For ease of comparison,

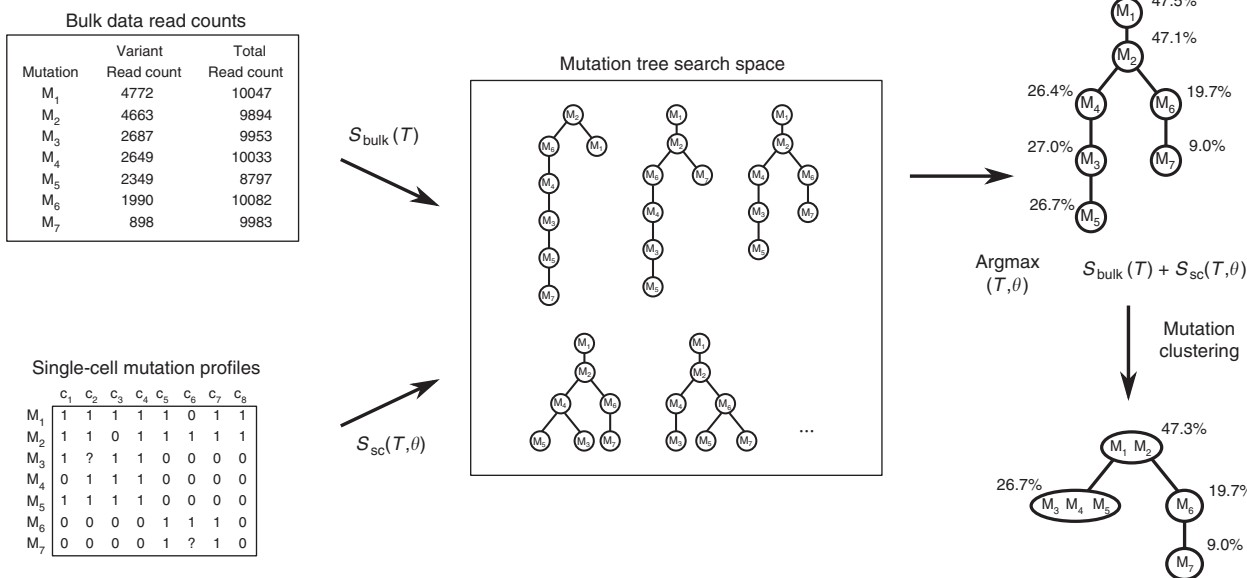

**Fig. 2** B-SCITE uses a Markov chain Monte Carlo approach to search the space of candidate mutation histories. Each candidate tree is scored based on its joint fit to the single-cell and bulk DNA sequencing data. The bulk data consist of a high-coverage variant and total read counts for the mutated loci. The single-cell data consist of the observed mutation profiles of the sequenced cells. These single-cell profiles are characterised by high noise rates ($\theta$) that are learned from the SCS data along with the tree topology $T$. B-SCITE reports the tree with the highest joint score. This tree is a fully resolved mutation tree. To obtain a clonal tree, the linear tree parts can be clustered based on the variant allele frequencies of the bulk data

we followed the simulation strategy employed in the original ddClone publication (see Supplementary Methods for details).

Since B-SCITE infers tumour phylogenies, we additionally compared against the two tree inference methods, OncoNEM[29] and SCITE[28], which are working on single-cell data only.

All box plots used for presenting performance of the methods on the simulated data were generated using ggplot2[35], with its default settings and the data points overlaid.

**Comparisons of clustering accuracy.** Since ddClone does not provide any output related to the tree of tumour evolution, we used the V-measure of cluster assignments[36] and adjusted Rand index of cluster assignments in our comparisons. For B-SCITE, the clonal tree derived from the fully resolved maximum likelihood tree provides the mutation clusters. Namely, each clone $C$ defines a unique cluster consisting of all mutations appearing for the first time at $C$. OncoNEM also provides an option to cluster the data into subclones based on the inferred phylogeny, and is hence included in the comparison.

The results on simulations (Fig. 3 for 25 single cells and 10 clones) show that B-SCITE consistently outperforms both ddClone and OncoNEM. B-SCITE and ddClone are both robust to doublet contamination and distortion in the single-cell data sampling denoted by $\lambda$. The latter introduces a discrepancy between the genotype frequencies of the sequenced single cells and the subclone frequencies of the bulk tumour tissue, where larger values of $\lambda$ indicate better agreement between the frequencies (for more details see Supplementary Methods).

OncoNEM, which only utilises single-cell data, improves as the sampling of single cells more closely reflects the bulk tumour composition (as $\lambda$ increases). Even for highly distorted data, OncoNEM performs a little better than ddClone. However, when simulating a smaller number of clones (Supplementary Fig. 1 with

six clones instead of ten), the gap between OncoNEM and ddClone increases while B-SCITE remains the best performer.

Increasing the number of cells from 25 to 50 and 100 has a marginal effect on the accuracy with 10 clones (Supplementary Fig. 2, although more of the simulated clones may also be observed with more cells). As the number of clones increases (Supplementary Figs. 3, 4), OncoNEM's performance decreases although it is aided by larger cell numbers, while ddClone's performance starts to degrade, as more cells allow more of the simulated clones to be observed. B-SCITE retains the best and most stable performance. A similar pattern is seen when computing the accuracy with the adjusted Rand index (Supplementary Figs. 5–7), which amplifies the differences between the methods.

The effect of allelic dropout and false negatives is relatively mild on B-SCITE (Supplementary Fig. 8) and has a more noticeable effect on ddClone and OncoNEM. A similar dependence on false negatives is seen with a highly elevated false-positive rate (Supplementary Fig. 9), and the false positives lead to a small but clear decrease in accuracy for B-SCITE. OncoNEM also suffers a slight loss in accuracy, while ddClone actually improves marginally with the higher error rate though still with the worst performance overall.

**Accuracy in inferring phylogenetic order of mutations.** In addition to clustering mutations into subclones, B-SCITE also infers the complete phylogenetic history of a tumour. We therefore compared B-SCITE with the single-cell phylogenetic methods OncoNEM and SCITE based on three different accuracy measures (the definition of each measure is available in the Supplementary Methods). Specifically, for SCITE, we chose the extended version with the doublet model[27] to make sure that any change in performance can be fully attributed to the additional data available to B-SCITE.

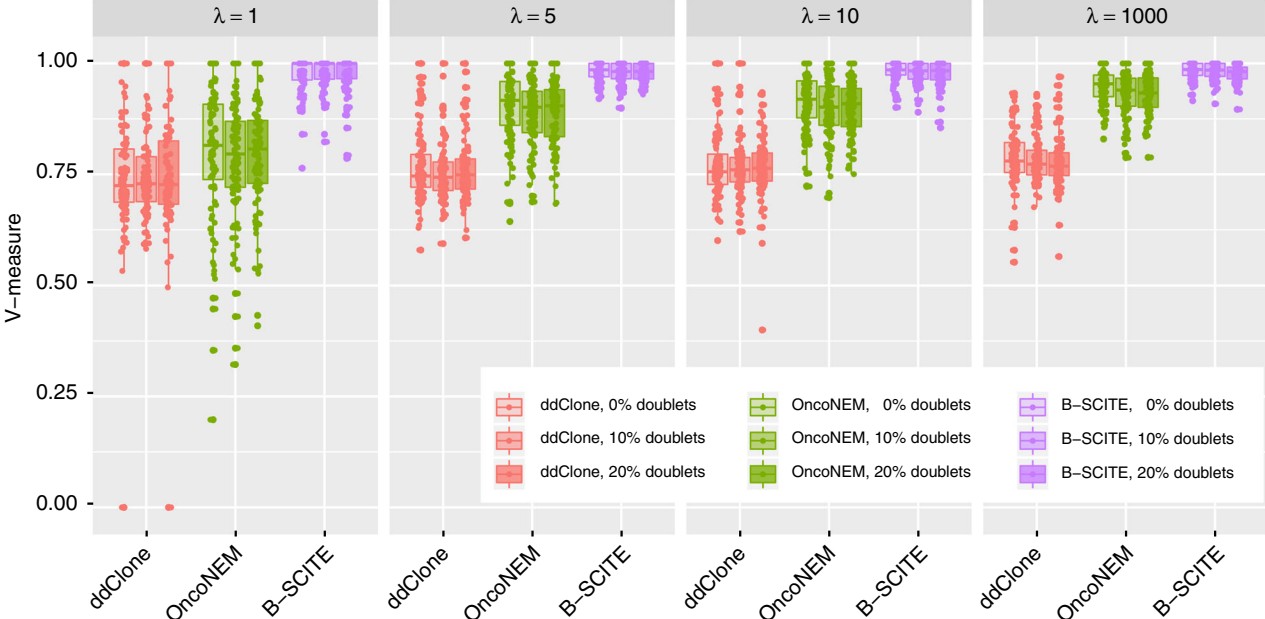

**Fig. 3** Accuracy of mutation clustering by ddClone, OncoNEM and B-SCITE for 100 simulated clonal trees with 10 nodes (clones) and 50 mutations. For the single-cell data, we drew 25 genotypes from each clonal tree for various values of parameter $\lambda$, which controls bias in sampling single cells from clones (large values of $\lambda$ indicate a small bias, where the probability of drawing a single cell from a given clone is usually close to its prevalence in the entire tumour cell population). We also added the following noise to the single-cell genotypes: false-positive rate $10^{-5}$, false-negative rate 0.2, missing (NA) rate 0.05 and doublet rates 0, 0.1 and 0.2. Bulk data coverage was set to 10,000, and variant read counts drawn from a binomial distribution. We obtained data sets from trees for each parameter combination. A more detailed description of the simulation data is given in Supplementary Methods. For the definition of V-measure see ref. [36]. Source data are provided as a Source Data file

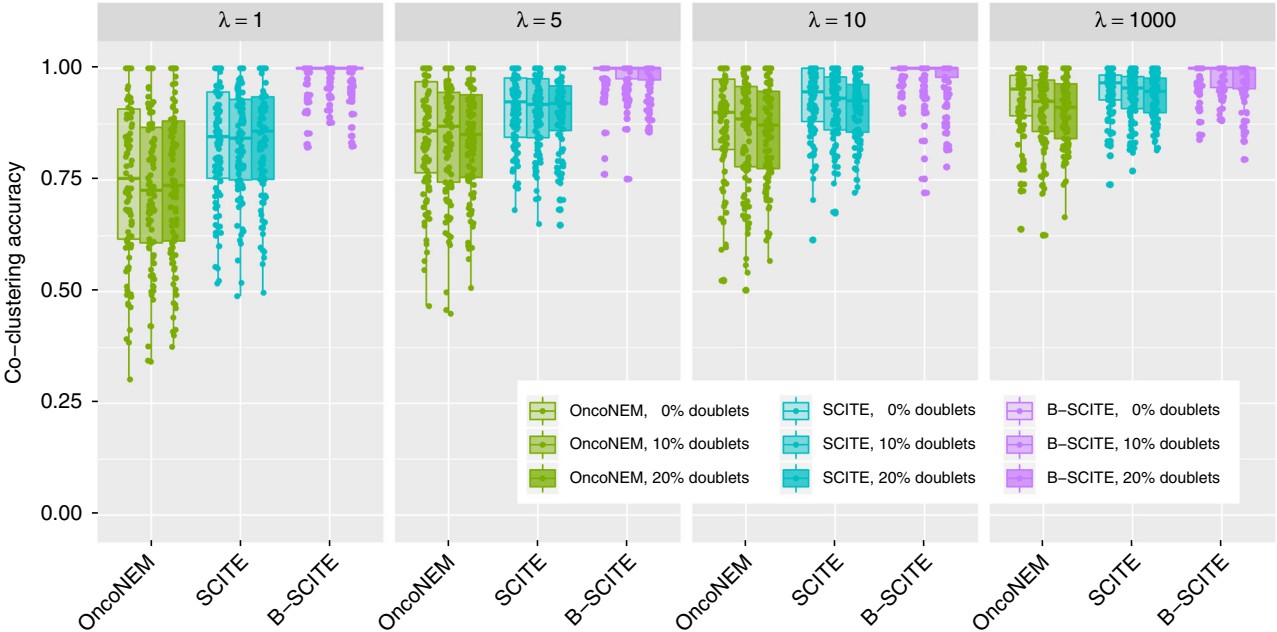

**Fig. 4** Comparison of phylogenetic inference for OncoNEM, SCITE and B-SCITE for 100 simulated clonal trees with 10 nodes (clones) and 50 mutations. For the single-cell data, we drew 25 genotypes from each clonal tree for various values of parameter $\lambda$, which controls bias in sampling single cells from clones (large values of $\lambda$ indicate a small bias, the where probability of drawing a single cell from a given clone is usually close to its prevalence in the entire tumour cell population). We also added the following noise to the single-cell genotypes: false-positive rate $10^{-5}$, false-negative rate 0.2, missing (NA) rate 0.05 and doublet rates 0, 0.1 and 0.2. Bulk data coverage was set to 10,000, and variant read counts drawn from a binomial distribution. We obtained data sets from trees for each parameter combination. A more detailed description of the simulation data and the definition of the co-clustering accuracy measure are given in Supplementary Methods. Source data are provided as a Source Data file

B-SCITE again has the best and most robust performance over the range of $\lambda$ (Fig. 4). The two single-cell methods improve, as the single-cell sampling approaches a better representation of the bulk frequencies, but never reaches the performance of B-SCITE. The apparent improvement for B-SCITE as $\lambda$ decreases is due to the smaller number of observed clones being included in the calculation of the tree accuracy.

Changing the number of clones or cells (Supplementary Figs. 10–13), we observe the same pattern with B-SCITE on top, and SCITE performing slightly better than OncoNEM. Similar behaviour is also observed for inferring the correct ancestry relationships (Supplementary Fig. 14). For mutations in separate lineages, both SCITE and B-SCITE perform well in correctly detecting the correct separations, while OncoNEM has a somewhat lower accuracy (Supplementary Fig. 15).

The effect of allelic dropout on the phylogenetic inference is again relatively mild on B-SCITE compared with SCITE and OncoNEM, while false positives play a more important role (Supplementary Figs. 16, 17) and reduce the accuracy of all tools by a small amount.

**Performance in the presence of CNA events**. Copy-number aberrations (CNAs) perturb the fraction of mutated reads in the affected regions, thereby shifting the observed VAFs. While the model underlying B-SCITE expects mutations to come from copy-number-neutral regions, it is not always possible to identify and discard all other mutations. VAFs from copy-number-altered regions are known to confound tree reconstruction and mutation ordering from bulk data only. By also utilising single-cell data, B-SCITE is quite robust to these effects, with only a small average decrease in the co-clustering accuracy, as the number of CNAs in the simulated data increases (Supplementary Fig. 18). For very-high-coverage data, where the bulk VAFs play a stronger role in the phylogenetic reconstruction, the effect is correspondingly

more pronounced (Supplementary Fig. 19). For the other accuracy measures, we see the same pattern for the two coverage levels (Supplementary Figs. 20–23).

**Performance in the presence of ISA violations**. In the tree inference (Methods), we employ the infinite-sites assumption (ISA), which states that mutations only occur once in the phylogeny and persist in descendent cells. Violations of this assumption however only have a very weak effect on the structural accuracy of the other mutations (Supplementary Fig. 24).

**Multiple bulk samples**. To assess B-SCITE's performance in settings where multiple bulk samples are available and to see whether additional bulk samples render single-cell data redundant, we simulated data with up to four bulk samples and compared B-SCITE with the bulk-only method PhyloWGS[18] (Supplementary Figs. 25–27). Expectedly, reconstruction quality of both tools improves, as the number of bulk samples increases. While B-SCITE generally outperforms PhyloWGS, it is possible to create settings that benefit the bulk-only method, namely a combination of high-quality and quantity bulk data with single-cell data that badly reflects the tumour composition (small $\lambda$). When looking at different settings for the bulk data, we observe more accurate reconstruction with multiple bulk samples at lower coverage as compared with a single bulk sample at high coverage. Finally, for any fixed number of bulk samples, we observe that the reconstruction quality improves with the number of available single cells.

**Application to real data**. To assess the performance of B-SCITE on real tumour data, we analysed the sequencing data of two patients with childhood leukaemia[37], one triple-negative breast cancer patient[38] and two colorectal cancer patients with matched

liver metastases[39]. Discussion of the results is provided below and all details of data pre-processing can be found in Supplementary Methods.

**Acute lymphoblastic leukaemia (ALL).** For both leukaemia patients, a bulk sample was sequenced together with a large number (>100) of single cells, for which targeted sequencing was performed using a personalised panel. This allows us to compare B-SCITE with methods relying solely on single-cell or bulk-sequencing data. For our comparison, we chose CTPsingle[20] for the bulk-only approach and SCITE[28] for single-cell-based inference.

For patient 1, 20 mutations were detected sequencing one bulk sample and 111 single cells. The phylogenies inferred by B-SCITE and its two competitors are depicted in Fig. 5. CTPsingle, the approach based on bulk samples, finds two trees compatible with the observed frequencies. Both trees cluster the 20 mutations in five subclones and are either completely linear or have a single mutation in a separate branch. Using the SCS data of the same patient, SCITE detects an early branching event that splits up some of the subclones inferred by CTPsingle. Without knowledge of the ground truth tree, there is no certainty whether the branching reflects the true phylogeny. However, having data from such a large number of cells and the close location to the root, makes it highly likely that the branching is genuine. The same branching is also inferred by B-SCITE, which finds generally the same topology as SCITE, but arranges mutations differently in linear segments of the tree. As expected, having the additional information of VAFs for the individual mutations allows B-SCITE to find an ordering that is in better congruence with the observed mutation frequencies, which should be in decreasing order. As a consequence, the mutation ordering inferred by B-SCITE is also closer to the subclone clustering of CTPsingle, in the sense that mutations from the same cluster that are in the same branch tend to be closer together than in the SCITE tree. For this data set, we estimate the relative weight of the bulk data to be 0.81 times that of the single-cell information.

For patient 2, 16 mutations were detected sequencing one bulk sample and 115 single cells. CTPsingle reports a subclone clustering that is compatible with many phylogenies and some of them are depicted in Fig. 6a and Supplementary Fig. 28. SCITE and B-SCITE each infer a single tree (Fig. 6b, c). The general topologies of the two trees are again similar with different arrangement of mutations on linear segments. In particular, B-SCITE puts the mutations in *FAM105A* and *CMTM8* higher up in their branch, which better reflects their relatively high frequencies (44% and 42%). Notably, B-SCITE does not completely sort the mutations in this branch by frequency, as the mutation in *RRP8* is placed between two mutations with higher frequencies. This indicates that a strong signal for this placement is coming from the single-cell data. Apart from issues with mutation calling, a possible explanation could be a copy-number change affecting *RRP8*, which decreases its observed VAF. The bulk data play a stronger role for these data, with their estimated weight to be 1.55 times that of the single cell.

**Triple-negative breast cancer (TNBC).** For the triple-negative breast cancer patient, in the original study[38], copy-number profiling of 50 single cells was first performed. Next, 16 single tumour cells were whole-exome sequenced and hierarchical

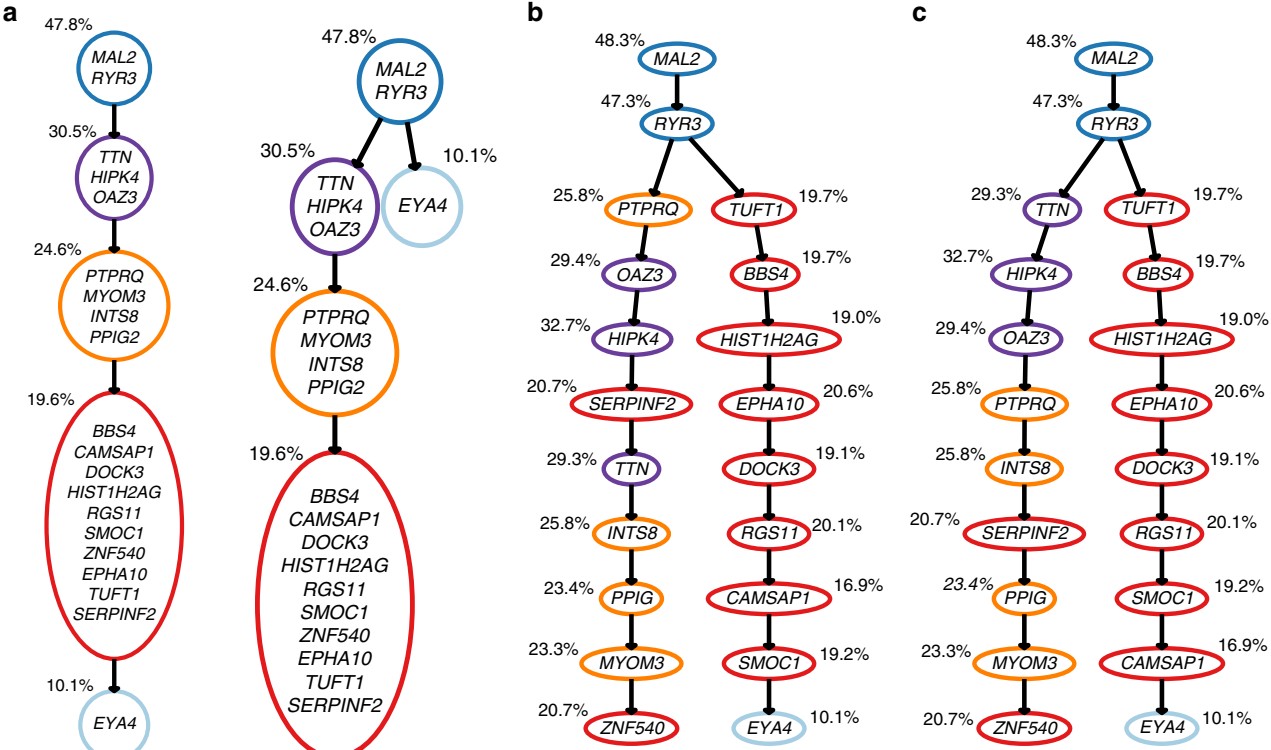

**Fig. 5** Mutation histories inferred from the bulk and single-cell sequencing data for a patient with childhood ALL (Patient 1 of the study in ref. [37]): **a** Clonal trees compatible with the bulk-sequencing data (inferred with CTPsingle[20]). Clones are annotated with the mean VAF of the mutations newly acquired by the clone. **b** Mutation tree inferred with SCITE[28] from the single-cell panel sequencing data of 111 cells. The colouring scheme follows from the clustering in the CTPsingle trees. Mutations are annotated with the VAFs observed in the bulk sample. **c** Mutation tree inferred with B-SCITE from the combined single-cell and bulk data. B-SCITE infers the same early branching event as SCITE, but finds a mutation ordering that is in better congruence with the bulk VAFs. B-SCITE mutation trees can be compressed to clonal trees (Supplementary Fig. 29)

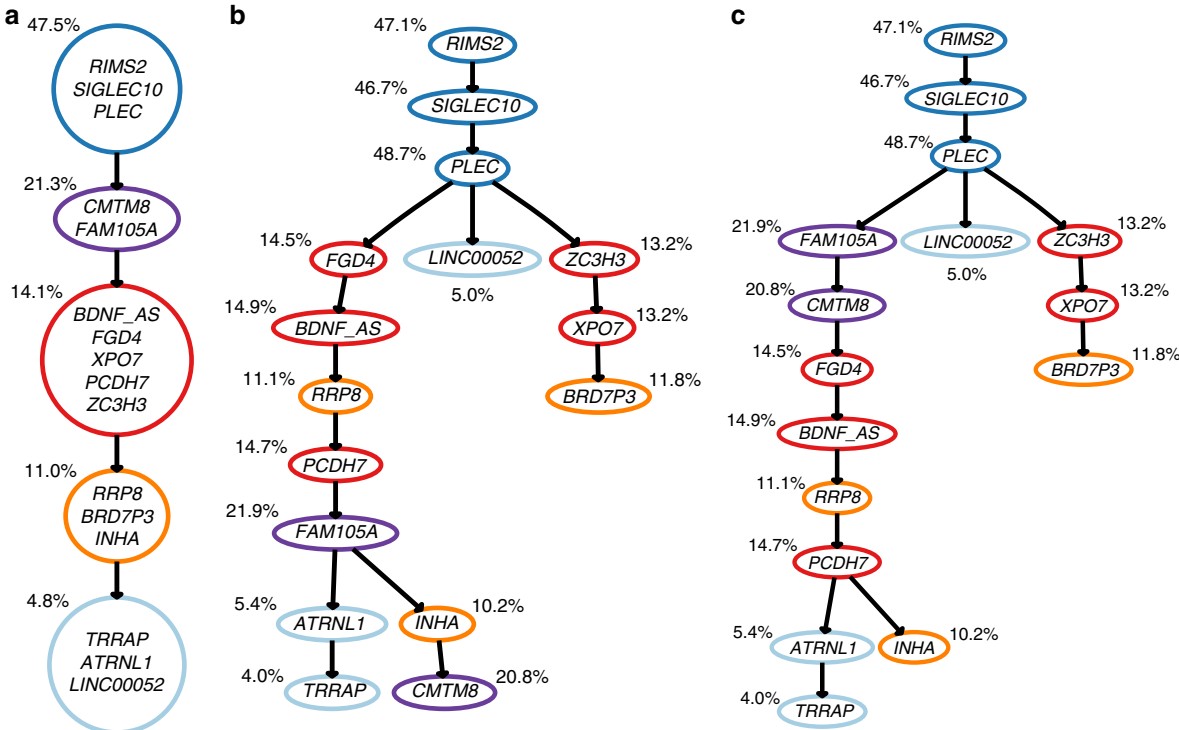

**Fig. 6** Mutation histories inferred from bulk and single-cell sequencing data for a patient with childhood ALL (Patient 2 of the study in ref. [37]): **a** one of the multiple clonal trees compatible with the bulk-sequencing data (inferred with CTPsingle[20]). Clones are annotated with the mean VAF of the mutations newly acquired by the clone. Several other compatible trees are depicted in Supplementary Fig. 28. **b** Mutation tree inferred with SCITE[28] from single-cell panel sequencing data of 115 cells. The colouring scheme follows from the clustering in the CTPsingle tree. Mutations are annotated with the VAFs observed in the bulk sample. **c** Mutation tree inferred with B-SCITE from the combined single-cell and bulk data. B-SCITE mutation trees can be compressed to clonal trees (Supplementary Fig. 29)

clustering of the cells was performed based on profiles of a large number (>500) of detected somatic SNVs, each present in more than one of the 16 cells (mutations detected in only one cell were discarded). A clonal tree was then manually reconstructed (Extended Data Fig. 6b in ref. [38]). Based on this tree, we provide a clonal tree for the 18 mutations selected in Fig. 7a. For each of these 18 mutations, the original study obtained bulk data read counts from targeted ultra-deep sequencing, with an approximate depth of coverage > 1,00,000×. Hence, we were able to run both, SCITE and B-SCITE, on this data set as well. Trees obtained by these tools are shown in Fig. 7b, c. Notably, B-SCITE infers a mutation tree, which is in a strong concordance with the clonal tree, and at the same time, in agreement with the variant allele frequencies obtained from bulk data. On the other hand, the tree inferred by SCITE has several differences compared with the clonal tree, including the placement of the mutation in gene *SYNE2* on a different branch than the *PPP2R1A* and *AURKA* mutations. Similarly, mutations reported to be clonal in ref. [38], namely mutations in genes *MAP2K7* and *NTRK1*, are placed on different branches in the SCITE tree, and the placement of the branch containing mutations in genes *CHRM5* and *TGFB2* is closer to the root. In this tree, there are also several inconsistencies with variant allele frequencies of mutations obtained from bulk data. With the high bulk coverage and the limited number of single cells, the bulk data have a much higher relative weight (≈2800 times) than the single-cell data. However, this does not mean that the latter have no effect on the tree structure. In fact, the bulk data favour a completely linear tree topology.

**Colorectal cancer.** We reanalysed the bulk and single-cell sequencing data of two cases of colorectal cancer with liver

metastases[39]. The interesting feature of these data sets is the availability of two bulk samples (primary tumour and metastasis) in addition to single-cell data. Unfortunately, the original study revealed in both patients the presence of aneuploid populations at both sites via flow sorting, which is not ideal for the use of B-SCITE due to CNA-based shifts of bulk VAFs.

In the study, flow-sorted cells were pooled and exome sequenced en bulk. We obtained the read counts from these experiments by processing the raw data from the SRA and used them as bulk data input. The original study also comprised targeted single-cell sequencing based on the T1000 cancer gene panel (mean depth 137× at average coverage of 0.92). To run B-SCITE, we focused on the mutations detected via single-cell panel sequencing, but discarded mutations with insufficient coverage in the bulk samples. In addition, we removed cells where no mutations were detected. These filtering steps left 12 mutations and 72 single cells for CRC1 (CO5), and 25 mutations and 86 single cells for CRC2 (CO8). The trees inferred by B-SCITE are displayed in Fig. 8.

For CO5, we observe an almost linear tree structure. For the primary tumour sample, VAFs are roughly decreasing, while we observe a bigger fluctuation for the metastasis sample. This may be a direct consequence of the higher aneuploidy observed in the liver metastatis compared with the primary site[39]. Compared with the targeted sequencing of the previous case, bulk-sequencing depth is relatively low here (with an average coverage depth of 69.25×). This is also reflected in a lower relative weight for the bulk data (≈1.75).

For CO8, we observe a more branched structure. Notably, there are two completely separate lineages, one of which is only detected in the single-cell data. In the other branch, early

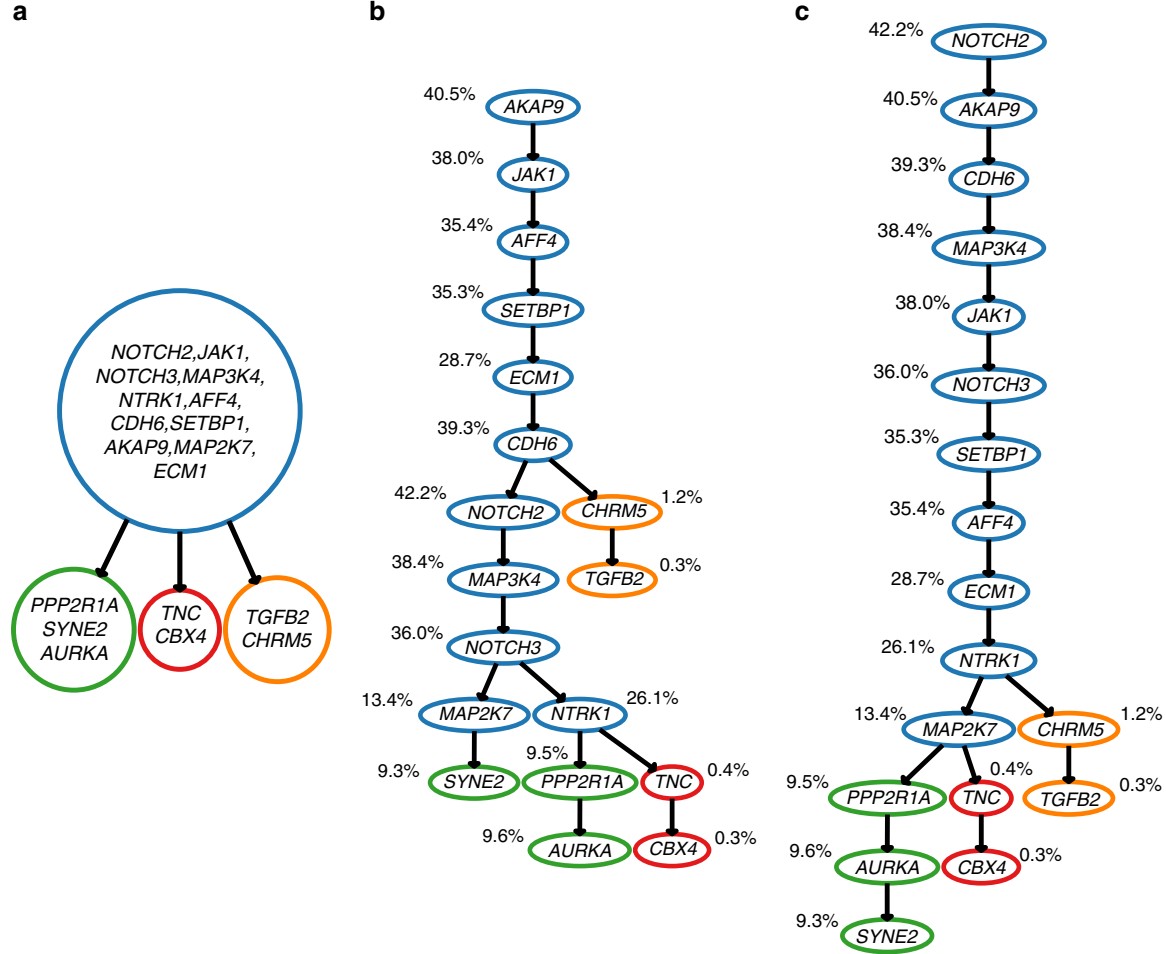

**Fig. 7** Mutation histories inferred from bulk (targeted sequencing with the average depth of coverage ~100,000×) and single-cell (16 cells) data for a patient with triple-negative breast cancer from ref. [38]: **a** clonal tree implied by hierarchical clustering from the original study[38]. Clustering was performed based on the profiles of several hundreds of mutations and four populations of cancerous cells were detected. In the same study, a subset of mutations was selected from each cell population and each of them is listed in the label of the corresponding population. **b** Mutation tree inferred with SCITE[28]. The colouring scheme follows the colouring used in the clonal tree. Mutations are annotated with the VAFs observed in the bulk sample. **c** Mutation tree inferred with B-SCITE from the combined single-cell and bulk data. B-SCITE infers highly similar branchings as in the clonal tree and also finds a mutation ordering that is in congruence with the bulk VAFs. B-SCITE mutation trees can be compressed to clonal trees (Supplementary Fig. 29)

mutations all have VAFs >50%, indicating the presence of CNAs. Fluctuation in VAFs in this tree part may indicate complex copy-number events, but at the given sequencing depth (average of 105.76×) may also be attributed to read count variance. The relative weight on the bulk data is 2.96 compared with the single-cell data.

The lower branches are generally in good agreement with decreasing bulk VAFs and show a clear distinction between the primary and metastatic sample. Interestingly, the original study reported two separate metastatic seeding events based on a tree topology inferred from single-cell data alone[39]. In particular, it places *FUS* (VAF 29.1% in the metastasis sample) in the second metastatic lineage, while other mutations with similar VAFs in the metastasis sample (*LAMB4* 27.6%, *F8* 30.0% and *SPEN* 25.1%) are placed in the first metastatic branch. This separation strongly violates the 'sum rule' for bulk VAFs and is likely the reason why B-SCITE chooses a topology which places these mutations into a single lineage. To better understand this discrepancy, we checked the single-cell genotype matrices inferred in the original study (Supplementary Fig. 7)[39] and found that the separate placement of *FUS* requires that *FUS* mutation calls are explained away as false positives in ten metastatic cells (MA_44, MA_42,

MA_41, MA_48, MA_29, MA_45, MA_90, MA_91, MA_33 and MA_35). Another ambiguous mutation is *ATP7B* (3.1% metastasis VAF), which the single-cell tree places above the metastatic mutation *FUS*, while B-SCITE places it in the primary branch. While in better congruence with the bulk data, the placement chosen by B-SCITE comes at the cost of an increased number of false-positive calls for *ATP7B*.

In general, there appears to be no tree topology that jointly explains all aspects of the single-cell data, let alone both data types. A possible explanation could be the presence of complex events that are not covered by our tree model, such as mutation loss or recurrent mutations. The absence of a well-fitting tree topology leads to two unwanted effects, both of which we observe in this data set. First, the inferred tree topology can be strongly influenced by the mutation choice. In the present data, lowering the read count cut-off (to 5 instead of 20) and thereby discarding fewer mutations is sufficient to drive *FUS* into a separate metastatic branch. Second, for a fixed set of mutations, lower branches in the tree can become unstable in the sense that multiple branching variations obtain highly similar scores. Having such a 'flat landscape' in high-scoring parts of the tree search space makes it difficult to find a globally optimal tree. This

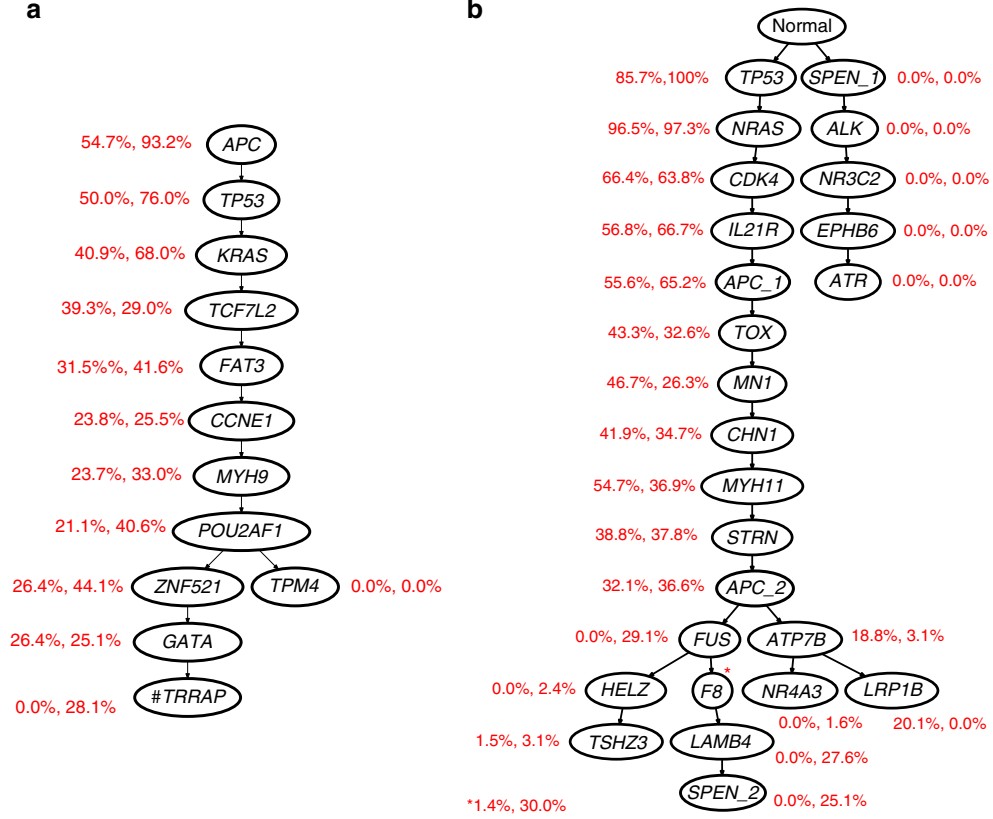

**Fig. 8** B-SCITE mutation histories for two cases of colorectal cancer with liver metastases[39]. **a** CRC1 (CO5) and (**b**) CRC2 (CO8). Mutations are annotated with the VAFs of the two bulk samples (left: primary tumour, right: liver metastasis). Bulk data come from whole-exome sequencing (at 75.5× depth and 97.33% coverage) of pooled flow-sorted single cells (for aneuploidy detection). Single-cell data come from targeted panel sequencing (T1000 cancer gene panel) at mean depth 137× with an average coverage of 0.92. B-SCITE was run on mutations detected in the panel sequencing that had a minimum read depth of 20 in the bulk samples. Cells without any of the remaining mutations were discarded for being non-informative in the tree inference. These filtering steps left 12 mutations and 72 single cells for CRC1 (CO5), and 25 mutations and 86 single cells for CRC2 (CO8). Fluctuating VAFs are likely due to a combination of CNAs and artefacts from limited sequencing depth in the whole-exome sequencing data. Note that in CRC2, *APC* and *SPEN* have each two independent point mutations denoted as (*APC_1*, *APC_2*, *SPEN_1* and *SPEN_2*)

matches our observation that this data set took much longer MCMC chains to consistently find the best tree than normally expected for data sets of this size.

## Discussion

Recent advances in sequencing technologies allow large-scale bulk and single-cell sequencing of tumour samples. The resulting data are invaluable for understanding the evolutionary history and subclonal composition of individual tumours and addressing the issue of treatment failure due to resistant cell populations. The bottleneck to fully leveraging the joint strength of both data types is the lack of specialised software which integrates single-cell and bulk-sequencing data in a joint inference scheme. Prior to this work, only a single tool (ddClone) has been available, where subclone inference based on bulk-sequencing data is informed by single-cell genotypes, but no integrative tool has been published for phylogeny inference. To fill this gap, we have developed B-SCITE, the first approach for inferring tumour phylogenies and subclonal compositions from combined single-cell and bulk-sequencing data. B-SCITE uses a joint likelihood model to integrate both data types and performs a probabilistic search to find the best combination of a fully resolved mutation history and values for the model parameters. Extensive simulation studies show that B-SCITE systematically outperforms competing single-cell-based approaches, thereby indicating that bulk data make

a valuable contribution to the inference. The quantity of the performance gain depends on the degree of sampling distortion between the single-cell samples and the composition of the bulk tumour. However, even in cases where the single-cell data very well reflect the tumour composition, B-SCITE outperforms its competitors.

Since the identification of the correct copy-number states of mutated sites can be a challenge, we performed additional tests with bulk VAFs that one could expect to arise from non-diploid regions. While B-SCITE is designed and works best for heterozygous mutations at copy-number-neutral sites, it still shows a robust performance when a subset of the mutation sites is affected by CNAs.

To compare B-SCITE with ddClone, the only other tool combining single-cell and bulk-sequencing data, we obtained subclones from B-SCITE's fully resolved mutation histories by performing a local mutation clustering. Experiments on a comprehensive set of simulated data sets showed that B-SCITE systematically outperforms ddClone, suggesting that subclone inference benefits from taking the underlying phylogeny into account.

In addition, we explored the usefulness of combining bulk and single-cell data in the analysis of real tumours. Looking at the data from two patients with childhood leukaemia from ref. [37], where a large number (>100) of single cells were sequenced, we find that B-SCITE and the single-cell-only approach SCITE

infer very similar branchings in the tumour phylogenies, while B-SCITE shows an improved ability in reconciling the temporal ordering of mutations with their VAFs. In cases where B-SCITE refrains from ordering mutations by decreasing VAFs, we suspect the presence of a copy-number change, whose signal was over-ridden by a strong contrary signal in the single-cell data. This gives further evidence that B-SCITE is to some extent robust to data deviating from the assumption that all mutations are heterozygous and at copy-number-neutral sites.

We also tested B-SCITE on a triple-negative breast cancer patient from ref. [38], from which 16 single cells were sequenced. In contrast to SCITE, B-SCITE infers a tree, which is in high concordance with the expert-generated tree for the 18 mutations pre-selected in the original study. These results suggest potential advantage of using B-SCITE over the existing methods in clinical settings, where a targeted subset of cancer-relevant genes are sequenced.

## Methods

**Tree models of tumour evolution**. A clonal tree $T$, depicting subclonal tumour evolution in a cancer patient with a set of detected somatic mutations $\mathbf{M} = \{M_1, M_2,..., M_n\}$, is a labelled rooted tree with node set $V(T) = \{v_1, v_2,..., v_s, v_{s+1}\}$. Its root $v_{s+1}$ represents the population of healthy cells of the patient, and the other nodes represent $s$ tumour subclonal populations (subclones) of the same individual, which are all genetically different. The mutational label associated with a node $v_i$, denoted as $L_T(v_i)$, is a subset of $\mathbf{M}$ and represents the set of mutations acquired at subclone $v_i$. In other words, for each non-root node $v_i$, $L_T(v_i)$ represents the set of all mutations present at subclone $v_i$, but absent from subclone $v_j$, where $v_j$ is parent of $v_i$ in $T$. We assume that the population of healthy cells does not harbour any mutation and therefore set $L_T(v_{s+1}) = \varnothing$. As subclones are all genetically different, we have that $L_T(v) \neq \varnothing$ for each $v \in \{v_1, \ldots v_s\}$. Furthermore, assuming that we have $h$ different bulk samples of the analysed tumour, we also add a frequency label to each node $v_i \in V(T)$. The frequency label of node $v_i$ is represented as a vector $(\phi_{i1}, \phi_{i2},..., \phi_{ih})$, where $\phi_{ij}$ denotes the relative frequency of cell population $v_i$ in bulk sample $j$.

In this work, we assume that each mutation was acquired exactly once by one subclone and then passed on to all its descendants (infinite-sites assumption). Consequently, $L_T(v_i) \cap L_T(v_j) = \varnothing$ for all pairs $(v_i, v_j)$ (mutations are acquired only once) and $\bigcup_{i=1}^{s} L_T(v_i) = \mathbf{M}$ (each of the observed mutations emerged in some of the subclones). As we assume that mutations are never lost, the set of mutations present in a subclone $v_i$ consists of all mutations acquired along the path from the root to $v_i$.

The topology of clonal tree $T$ can be represented by an $s \times s$ ancestor matrix $A_T$ defined as follows (we assume 1-based indexing of matrix entries):

$$A_T[i, j] = \begin{cases} 1, & \text{if } v_i \text{ is an ancestor of } v_j \text{ in } T, \text{ or } i = j \\ 0, & \text{otherwise.} \end{cases} \quad (1)$$

Below, we also allow the use of $A_T[v_i, v_j]$, which is defined to be equal to $A_T[i, j]$.

For convenience of notation, we also introduce function $N_T: \mathbf{M} \rightarrow V(T)$ with $N_T(M_i)$ defined as the node of the first occurrence of mutation $M_i$ in clonal tree $T$. In other words, $N_T(M_i) = v_j$ if and only if $M_i \in L_T(v_j)$.

For tree inference based on single-cell data, the mutation tree model that does not cluster mutations into subclones is more natural. The mutation tree can be defined as a special case of a clonal tree where $s = n$. This obviously implies that $|L_T(v_i)| = 1$ for each non-root node $v_i$ from $V(T)$. Furthermore, for each such node and without loss of generality, we may assume that $L_T(v_i) = M_i$.

**Input data**. We assume that a set $\mathbf{M} = \{M_1, M_2,..., M_n\}$ of heterozygous somatic single-nucleotide variants from diploid regions of the genome is given. These mutations were observed in a tumour via sequencing of $h$ bulk samples $\mathscr{B}_1, \mathscr{B}_2,..., \mathscr{B}_h$, and $m$ single-cell samples $\mathscr{C}_1, \mathscr{C}_2, \ldots, \mathscr{C}_m$.

For each mutation $M_i$ and each bulk sample $\mathscr{B}_j$, the bulk sequencing provided the number of variant and total reads spanning the genomic position of $M_i$ in $\mathscr{B}_j$, denoted, respectively, by $r_{ij}$ and $t_{ij}$. SCS provided the observed mutation profiles of cells $\mathscr{C}_1, \mathscr{C}_2, \ldots, \mathscr{C}_m$ as the column vectors of a mutation matrix $D_{n \times m}$ defined as

$$D[i, j] = \begin{cases} 0, & \text{if } M_i \text{ is reported to be absent from } \mathscr{C}_j \\ 1, & \text{if } M_i \text{ is reported to be present in } \mathscr{C}_j \\ NA, & \text{if status of } M_i \text{ in } \mathscr{C}_j \text{ is not known (missing entry).} \end{cases} \quad (2)$$

**Tree scoring based on bulk-sequencing data**. Assume that we are given a mutation tree $T$ over $n = s$ mutations, whose $(s + 1)$ nodes represent the set of cellular subpopulations in the analysed tumour. For each bulk sample $\mathscr{B}_j$, our goal is to assign non-negative real values $\Phi_j = \{\phi_{1j}, \phi_{2j},..., \phi_{(s+1)j}\}$ to the nodes

of $T$ such that

$$\phi_{1j} + \phi_{2j} + \ldots + \phi_{(s+1)j} = 1 \quad (3)$$

and the likelihood of bulk-sequencing data (defined below) be maximised. Intuitively, $\phi_{ij}$ represents the inferred fraction of the cellular population $v_i$ in the sample $\mathscr{B}_j$.

Consider an arbitrary mutation $M_i$. By the assumption made above, $M_i$ occurs for the first time at node $v_i$ and is present at $v_i$ and all of its descendants (i.e., nodes $v_j$ such that $A_T[i, j] = 1$). This implies that the inferred fraction $y_{ij}$ of cells harbouring mutation $M_i$ in sample $\mathscr{B}_j$ is given by the formula

$$y_{ij} = \sum_{l=1}^{s} A_T[i, l] \cdot \varphi_{lj}. \quad (4)$$

Alongside the tree constraints in Eqs. (3) and (4), we introduce a likelihood model for the bulk-sequencing data to allow us to combine it with the single-cell measurements.

Consider a genomic position $P$ in a diploid genome region that has a (sub-)clonal heterozygous mutation $M \in \mathbf{M}$ in a tumour. Let $y$ be the fraction of cells harbouring $M$ in an arbitrary bulk sample. Assume that we sequence this bulk sample and obtain a total of $t$ reads that span $P$, of which $r$ supports the variant $M$. Based on the composition of the considered bulk sample, the probability that a read supports $M$ is $\frac{y}{2}$ (since $M$ is heterozygous and from a diploid region). We assume that the sampled number of variant reads $r$ follows a binomial distribution with parameters $t$ (number of trials) and $\frac{y}{2}$ (success probability). For high-coverage $t$, the binomial distribution can be approximated by the Gaussian distribution with mean $\mu = t \cdot \frac{y}{2}$ and standard deviation $\sigma = \sqrt{t \cdot \frac{y}{2} \cdot \left(1 - \frac{y}{2}\right)}$. The logarithm of the probability density is

$$\log \frac{1}{\sqrt{2\pi}} + \log \frac{1}{\sqrt{t \cdot \frac{y}{2} \cdot \left(1 - \frac{y}{2}\right)}} - \frac{t}{8 \cdot \frac{y}{2} \cdot \left(1 - \frac{y}{2}\right)} \cdot (z - y)^2 \quad (5)$$

where $z = \frac{2r}{t}$, which represents the bulk-sequencing data-based fraction of cells harbouring mutation $M$, based on the assumption that $M$ is from a diploid region. Full details of derivation of Eq. (5) are provided in the Supplementary Methods.

For a given assignment of values $y_{ij}$, after discarding constant terms, the log-likelihood of the entire bulk data is then

$$\sum_{i=1}^{n} \sum_{j=1}^{h} \left[ \log \frac{1}{\sqrt{t_{ij} \cdot \frac{y_{ij}}{2} \cdot \left(1 - \frac{y_{ij}}{2}\right)}} - \frac{t_{ij}}{8 \cdot \frac{y_{ij}}{2} \cdot \left(1 - \frac{y_{ij}}{2}\right)} \cdot \left(z_{ij} - y_{ij}\right)^2 \right]. \quad (6)$$

Our goal is to maximise the likelihood over the latent variables $y_{ij}$, under the constraints Eqs. (3) and (4) imposed by the tree topology. To make this maximisation problem tractable by existing solvers, we use bulk data-derived frequencies $z_{ij} = \frac{2r_{ij}}{t_{ij}}$ to approximate the standard deviations

$$\sigma_{ij} = \sqrt{t_{ij} \cdot \frac{y_{ij}}{2} \cdot \left(1 - \frac{y_{ij}}{2}\right)} \approx \sqrt{t_{ij} \cdot \frac{z_{ij}}{2} \cdot \left(1 - \frac{z_{ij}}{2}\right)}$$

so that the log terms in Eq. (6) become constant and can be removed from the optimisation, while quadratic term coefficients become constants. Our problem then transforms to maximising the likelihood over the underlying frequencies. We therefore define the score of the bulk data to be

$$S_{\text{bulk}}(T) = \max_{y_{11}, y_{12}, \ldots, y_{(s+1)h}} \sum_{i=1}^{n} \sum_{j=1}^{h} \frac{-t_{ij}}{8 \cdot \frac{z_{ij}}{2} \cdot \left(1 - \frac{z_{ij}}{2}\right)} \cdot \left(z_{ij} - y_{ij}\right)^2,$$

subject to the restrictions imposed in Eq. (3) and Eq. (4), where the sum involves the weighted quadratic terms from the Gaussian approximation. Obtaining the optimal values for $y_{ij}$ represents an instance of a Quadratic Program (QP), which is readily solvable by the existing commercial and free QP solvers. For our purposes, we have used IBM ILOG CPLEX Optimization Studio V12.5.

**Tree scoring based on single-cell data**. For the tree scoring based on single-cell data, we need to assess how well the observed mutation states of the single cells match the subclones defined by $T$. Due to noise in the mutation matrix $D$, the single cells will likely fit to none of the $s + 1$ cell populations defined by $T$ perfectly, even if the tree represents the true mutation history of the tumour. We account for this by using the probabilistic error model introduced in ref. [28]: let the vector $\boldsymbol{\sigma} = (\sigma_1, \sigma_2,..., \sigma_m)$ define the attachments of the single cells to $T$, such that $c_j$, the single cell corresponding to column $j$ in $D$, attaches to $v_{\sigma_j}$. Then we expect $c_j$ to have the mutations assigned to the nodes $v_i$ that belong to the path from root to $v_{\sigma_j}$ (i.e., all nodes $v_i$ such that $A_T[i, \sigma_j] = 1$). The observational probabilities are then

$$\begin{aligned} P(D[i, j] = 1 | A_T[i, \sigma_j] = 0) &= \alpha \\ P(D[i, j] = 0 | A_T[i, \sigma_j] = 0) &= 1 - \alpha \\ P(D[i, j] = 0 | A_T[i, \sigma_j] = 1) &= \beta \\ P(D[i, j] = 1 | A_T[i, \sigma_j] = 1) &= 1 - \beta, \end{aligned} \quad (7)$$

where $\alpha$ denotes the probability of observing a false positive and $\beta$ denotes the probability of observing a false negative. The two error rates are summarised as $\boldsymbol{\theta} = (\alpha, \beta)$ in the following. By setting the probability of having missing observations ($D[i,j] = \mathrm{NA}$) to 1 independent of the true state, they are neutral and do not contribute to the tree scoring

$$P(D[i,j] = \mathrm{NA}|A_T[i,\boldsymbol{\sigma}_j] = 0) = 1$$
$$P(D[i,j] = \mathrm{NA}|A_T[i,\boldsymbol{\sigma}_j] = 1) = 1. \quad (8)$$

Assuming that the observational errors are independent of each other, the likelihood of a given mutation tree $T$, sample attachment vector $\boldsymbol{\sigma}$ and $\boldsymbol{\theta}$ is then

$$P(D|T,\boldsymbol{\theta},\boldsymbol{\sigma}) = \prod_{i=1}^{n}\prod_{j=1}^{m} P\Big(D[i,j]|A_T[i,\boldsymbol{\sigma}_j]\Big). \quad (9)$$

In the following, we marginalise out the sample attachments to focus on the mutation tree as the informative part of the mutation history, which is also more robust against noise than the location of individual samples in the tree. This gives us

$$P(D|T,\boldsymbol{\theta}) = \sum_{\boldsymbol{\sigma}} P(D|T,\boldsymbol{\theta},\boldsymbol{\sigma})P(T,\boldsymbol{\theta},\boldsymbol{\sigma}). \quad (10)$$

In practice, SCS data are often contaminated with doublet samples. Therefore, we treat each sample as a weighted mixture of a singlet and doublet sample[27]. The marginalised likelihood then becomes

$$P'(D|T,\boldsymbol{\theta}) = \max_{0 \leq \delta \leq 1}\Bigg\{\prod_{j=1}^{m}\Bigg[(1-\delta)\sum_{\boldsymbol{\sigma}_j}P(D_j|T,\boldsymbol{\theta},\boldsymbol{\sigma}_j)P(\boldsymbol{\sigma}_j|T,\boldsymbol{\theta})$$
$$+ \delta\sum_{\boldsymbol{\sigma}_j}\sum_{\boldsymbol{\sigma}_j'}P(D_j|T,\boldsymbol{\theta},\boldsymbol{\sigma}_j,\boldsymbol{\sigma}_j')P(\boldsymbol{\sigma}_j,\boldsymbol{\sigma}_j'|T,\boldsymbol{\theta})\Bigg]\Bigg\} \quad (11)$$

where $\delta$ is the probability of a sample being a doublet and $D_j$ is the $j$th column of $D$, which represents the observed mutation states of cell $j$. The double sum over the attachment points $\boldsymbol{\sigma}_j$ and $\boldsymbol{\sigma}_j'$ creates all combinations of attachment pairs. This can be efficiently computed in time $O(mn^2)$[27]. Finally to obtain a single-cell-based tree score that is on a comparable scale to the bulk score, we take the log of the marginalised likelihood

$$S_{\mathrm{sc}}(T,\boldsymbol{\theta}) = \log(P'(D|T,\boldsymbol{\theta})).$$

**Combined B-SCITE approach**. To measure how well a candidate mutation tree $T$ fits a combination of bulk and single-cell measurements, we use the joint log-likelihood score defined as

$$S_{\mathrm{joint}}(T,\boldsymbol{\theta}) = S_{\mathrm{bulk}}(T) + S_{\mathrm{sc}}(T,\boldsymbol{\theta}).$$

Our goal is to find a combination $(T,\theta)^*$ that maximises the above score:

$$(T,\theta)^* = \underset{(\mathbf{T},\theta)}{\mathbf{argmax}}\, \mathbf{S}_{\mathrm{joint}}(\mathbf{T},\theta).$$

The number of possible mutation trees is too large to allow for an exhaustive search. Therefore, we use the Markov chain Monte Carlo approach introduced in ref. [28] to search the joint $(T, \theta)$ space. In each step, it proposes a new state, which has either a new mutation tree $T'$ or a new parameter $\boldsymbol{\theta}'$. The proposal probability $q(T', \boldsymbol{\theta}'|T, \boldsymbol{\theta})$ is determined by the neighbourhood size of the respective move type. A new tree $T'$ is obtained from $T$ by (i) pruning and re-attaching of a subtree, by (ii) swapping two subtrees or by (iii) the exchange of the labels of two nodes (see ref. [28] for more details). The key difference to SCITE is that the probability of accepting a new state is now depending not only on the observed single-cell mutation profiles but also on the bulk VAFs. Hence, the acceptance probability becomes

$$\min\left\{1, \frac{q(T,\boldsymbol{\theta}|T',\boldsymbol{\theta}')\cdot e^{S_{\mathrm{joint}}(T',\theta')}}{q(T',\boldsymbol{\theta}'|T,\boldsymbol{\theta})\cdot e^{S_{\mathrm{joint}}(T,\boldsymbol{\theta})}}\right\}.$$

Note that $S_{\mathrm{bulk}}(T)$ does not depend on $\boldsymbol{\theta}$. Therefore, it needs not to be recomputed to obtain $S_{\mathrm{joint}}(T, \theta')$ after a $\theta$-move.

**Compression of mutation trees into clonal trees**. To compare the mutation tree $T$ inferred by B-SCITE with clonal trees inferred in OncoNEM or mutational clusters inferred in ddClone, we first perform clustering of mutations in $T$ in order to identify clones. Mutations placed on different branches in $T$ are expected to evolve in divergent clones; hence, we assume that they are not clustered together into the same clone. Furthermore, each of the branching nodes of $T$ (i.e., nodes having more than one child) represent the emergence of new clonal populations in the subtrees formed by its descendants. This motivated us to first identify all linear chains in $T$, which do not contain any branching node (except for the end nodes). Then, for each such chain, we discard its end node closest to the root and cluster the remaining mutations along the linear chain.

For the clustering, we employ a 1D Gaussian mixture model for the cellular frequencies $y_i$ of the mutations in the chain. The standard deviation of each component is fixed by its mean through the binomial approximation according

to Eq. (5). Therefore, only the mean of each component and the assignment of mutations to components need to be inferred, for which we employ the EM algorithm. The optimal number of mixture components is selected according to the Akaike information criterion.

As clustering is performed based on the inferred mutation frequencies, which decrease down the chain, only subsets of consecutive mutations can get clustered together. For each cluster of the mutations obtained in the clustering step, we merge their corresponding nodes into a single node and assign all mutations from the cluster to this node. The whole procedure results in the clonal tree of tumour evolution.

With multiple bulk samples, the mixture components and the assignment of mutations to them is shared across the samples, and we simply allow each component to have a different mean for each sample.

**Estimating the relative contribution of bulk and SCS data**. To estimate the range of the single-cell and bulk-based scores, we define a baseline tree, $\hat{T}$, with a single clone containing all mutations, which are then present in all single cells. The range of the log scores of each data type is then

$$S_{\mathrm{bulk}}(T^*) - S_{\mathrm{bulk}}(\hat{T}) \quad \text{and} \quad S_{\mathrm{sc}}(T^*) - S_{\mathrm{sc}}(\hat{T})$$

allowing us to define an estimate of the relative weight of the bulk data to the single cell as

$$\rho = \frac{S_{\mathrm{bulk}}(T^*) - S_{\mathrm{bulk}}(\hat{T})}{S_{\mathrm{sc}}(T^*) - S_{\mathrm{sc}}(\hat{T})}.$$

This estimate can inform the weighting parameter $\omega$ to balance the two data types:

$$\omega = \frac{\rho}{1 + \rho}.$$

**Reporting summary**. Further information on research design is available in the Nature Research Reporting Summary linked to this article.

## Data availability
The human sequencing data sets utilised in this study are available for download from the Sequence Read Archive with the accession numbers: SRP044380 (for ALL patients), SRA053195 (for TNBC patients) and SRP074289 (for CRC patients).

## Code availability
B-SCITE has been implemented in C++ and is freely available under a GNU General Public License v3.0 at https://github.com/smalikic/B-SCITE.

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

## Acknowledgements

S.M. was supported by a Vanier Canada Graduate Scholarship and NSERC CREATE (139277) fellowship. K.J. was supported by SystemsX.ch RTD Grant 2013/150 (http://www.systemsx.ch/). J.K. was supported by ERC Synergy Grant 609883 (http://erc.europa.eu/). S.C.S. was supported by NSF Grant CCF-1619081, NIH grant GM108348 and the Indiana University Grand Challenges Program, Precision Health Initiative. The authors would like to thank Jochen Singer and Farid Rashidi Mehrabadi for the help in mutation calling for TNBC data.

## Author contributions

S.M., K.J. and J.K. designed and implemented the method. S.M., K.J., J.K., C.S. and N.B. conceived the project and wrote the paper. All authors read and approved the final paper.

## Additional information

**Competing interests:** The authors declare no competing interests.

