## [Peer Review File · Nature Communications]

Reviewers' comments:

Reviewer #1 (Remarks to the Author):

This paper introduces the first computational approach for inferring tumor phylogeny from combined single-cell and bulk DNA sequencing data. The method is an extension of SCITE to include bulk variant allele frequencies (VAFs) for inferring tumor trees (inference is done assuming infinite sites model). It optimizes a joint likelihood function that has two components corresponding to bulk and single-cell sequencing data. The single-cell component is same as in SCITE. The bulk component represents an instance of Quadratic Programming (QP) that is solvable by QP solvers. The output is a mutation tree, and the authors present an algorithm to convert it to a clonal tree (by clustering nodes using Gaussian mixture model).

The method performs well compared to ddClone, OncoNEM and SCITE on simulated datasets. ddClone is the only other method that combines single-cell and bulk data for inference of tumor clones. The simulation setup takes into account different number of cells, different number of clones, and doublet rate. The number of mutations is either 50 or 100. The authors also have a simulation study where they account for the effect of copy number aberrations (CNA) on bulk VAF and show that the tree inference error does not suffer much. In simulation experiments, the method performs very well.

Comments:

- In their simulation studies, the authors used false positive (FP) rate of 10^{-5} . Given that most of their datasets involving 25 cells and 50 mutations, it is possible that no FP was present in the datasets. Even though FP rate in SCS datasets is on the order of $\sim 10^{-5}$, when looking at a mutation matrix, the FP probability α for simulation should be higher to introduce FPs in the datasets. This issue is highlighted for example in the papers that introduce the OncoNEM and SiFit methods. It is important that the authors look into this and report the amount of FPs in the simulated datasets.
- It is interesting to see that ddClone always performed worse compared to OncoNEM, which is completely opposite of what is reported in the ddClone paper. Same simulation settings were used as those in the ddClone paper. Could the reason be that the number of cells simulated in this study is smaller and ddClone is not getting enough information from small SCS datasets? Please elaborate.
- The authors used two "old" biological datasets (2014), and newer ones have become available. It might be good to run the method on newer datasets, such as the ones used in the SiFit paper.
- In most cases, the mutations in single-cell and bulk may differ. Sometimes, bulk data may have higher number of mutations being detected compared to targeted SCS data. Whether one should use the full set of bulk mutations or the set of mutations common to both is not clear. In particular, with respect to the proposed method/study, it is unclear what role (can this be quantified?) the SCS and bulk data play in the inference. That is, how much is the bulk data contributing to the inference, and how much is the SCS data contributing? Furthermore, the authors mention that both scores derived from SCS and bulk data are on the same scale, and that they weight them equally (the ω parameter). However, it is unclear why both scores are on the same scale. It would be good to report on the possible range of each score.
- The authors mentioned the affect of the number of single cells to the performance, i.e., the larger the number, the higher the affect it would have on the inference. However, the authors didn't report the effect of the sequencing depth of the bulk tissue, which could be done by a simulation study. A plot of the accuracy with the sequencing depth of bulk tissue might be beneficial to the community in informing the extent to which to sequence a patient.
- The authors report the best tree, i.e., the tree with the highest likelihood. However, other trees might shed light on the quality/significance of the top tree. For example, it would be good to report the tree with the second highest likelihood and contrast it to the best.
- The authors conducted all simulations under the infinite sites assumption. While their model/method

employ this assumption, could they simulate data that violate the assumption and study how the method performs on such data? This is important especially in light of the recent study by Beerenwinkel and colleagues demonstrating violation of this assumption in empirical datasets.
- Minor: First word on page 3: "complimentary" → "complementary".

Best regards,
Luay Nakhleh
(I sign all my reviews)

Reviewer #2 (Remarks to the Author):

The authors present B-SCITE, a method for phylogeny estimation from single-cell and bulk sequencing samples from the same tumor. The use of both data types has synergistic effects and B-SCITE is the first method to do so. Thus, the manuscript advances the state of the art and is of interest to the readership of this journal. Please see below for my comments.

1. Extend B-SCITE to support multiple bulk samples from the same tumor.

As mentioned by the authors a single bulk sample does not allow one to identify branching evolution. However, with multiple bulk samples such patterns start to become apparent. I encourage the authors to extend their model and algorithms to support multiple bulk samples from the same tumor. The use of multiple spatial/temporal bulk samples is becoming increasingly common place.

2. Improved/additional simulations.

Using simulated data, the authors compare B-SCITE against ddClone, OncoNEM and SCITE. To assess B-SCITE's improvement over the current state of the art, I would like to see a comparison against methods that only use bulk sequencing samples (a selection from CTPsingle, CITUP, AncesTree, PhyloWGS, etc.).

In addition, I would like to see simulations that assess the interplay between the number of bulk and single-cell samples. Moreover, the current simulations seem to assume a coverage of 10,000X. It would be good to assess B-SCITE's robustness in the case of lower coverage (e.g. WGS: 60X) and varying SCS false positive and false negative rates.

In addition to the V-measure, it would be a good idea to measure to clustering accuracy using more direct measures such as the Rand index.

The effect of copy-number aberrations seems to be only measured in terms of co-clustering accuracy. What happens with the other measures? Also, how is the difference defined (y-axis of Fig. A11)?

Finally, it is not completely clear how CNAs were simulated. (e_{i,f_i}) seem to denote changes in the variant and reference allele counts. What are the initial values? Does this number ever become negative?

Minor comments

* Last sentence of abstract: "... more realistic tumor histories." Please be more specific.

* Please be consistent in the notation of matrix entries ($A_T[i,j]$ vs $D_{\{i,j\}}$)

* Typo/awkward phrasing: "When $L_T(v_i)$ denotes the set of mutations..."

* Typo on page 7: subclons

* For completeness, it would be good to give a derivation of (5) in the supplement.

Reviewer #3 (Remarks to the Author):

This paper presents a novel inference method for cancer lineage trees that combines bulk sequencing data with single-cell sequencing data.

I am not equipped to go through all the math but I trust that it is methodologically sound. The overall architecture of the method is principled and represents an advance. I agree with the authors that this is the first method to combine single cell and bulk data to infer tumor clonal and lineage trees.

The authors make a clear case for the method, but the study does not go beyond methodological advances. The limited amount of analysis is confined to two patients, reflecting the difficulty to obtain, and therefore the rarity of, applicable data. I would be much more enthusiastic about the paper if novel data had been generated or if much more data had been analyzed, and application of the method had generated some insight.

One major concern that remains in any method that heavily relies on VAF to infer somatic lineage trees is the confounding influence of copy number variation on VAF. This is an issue that has not been solved and B-SCITE is as susceptible to it as any other method that relies on VAF. For example, VAF of close to 100% (e.g., MAL2 and RYR3 in Fig 5a) is almost certainly due to loss of the wild type chromosome (LOH); more insidiously, a VAF of 1/3 or 2/3 is almost certainly due to either the wild type or the mutant chromosome having duplicated, respectively. Unless these events are explicitly modeled, VAF-based inference cannot be considered reliable because it misinterprets genomic events (i.e., copy number changes) as differences in cellular representation, and thus gives predictably erroneous results. Given the prevalence of copy number changes in cancer, especially in solid tumors, this is a major issue for any such methods, including B-SCITE.

We would like to thank all reviewers for their helpful comments and constructive suggestions.

Response to Reviewer 1:

We thank Luay Nakhleh for his positive feedback and suggestions to improve our work.

*1. In their simulation studies, the authors used false positive (FP) rate of 10^{-5} . Given that most of their datasets involving 25 cells and 50 mutations, it is possible that no FP was present in the datasets. Even though FP rate in SCS datasets is on the order of $\sim 10^{-5}$, when looking at a mutation matrix, the FP probability α for simulation should be higher to introduce FPs in the datasets. This issue is highlighted for example in the papers that introduce the OncoNEM and SiFit methods. It is important that the authors look into this and report the amount of FPs in the simulated datasets.

We thank Luay for this comment and have added simulations with a FP rate of 1%. Although this is much higher than in real data, it allows us to see the effect on performance of such high values. Comparing Figures A8 and A9, there is a small but noticeable decrease in accuracy with the higher FP rate, which also affects OncoNEM so that B-SCITE still outperforms the alternatives.

*2. It is interesting to see that ddClone always performed worse compared to OncoNEM, which is completely opposite of what is reported in the ddClone paper. Same simulation settings were used as those in the ddClone paper. Could the reason be that the number of cells simulated in this study is smaller and ddClone is not getting enough information from small SCS datasets? Please elaborate.

While in our experiments we have used the same strategy for simulating distortion in single-cell data sampling as in the original publication of ddClone (Salehi et al. 2017), there are also some important differences between the two sets of experiments. First, we could not find missing entries simulated in Salehi et al. 2017, nor false positives (except false positives generated as a consequence of doublets noise). Second, the

rate of simulated false negatives and doublets used in Salehi et al. 2017 were both 0.30, whereas our simulations were generated using false negative rate of 0.20 and doublet rates of 0, 0.10 or 0.20. In order to test whether the difference in the results is a consequence of a possible glitch in our pipeline for running and comparing results of ddClone and OncoNEM, we also generated 100 simulations using the same parameters as in Figure 4 of Salehi et al. (number of cells = 50, number of mutations = 48, number of subclones = 10, single-cell data sampling distortion parameter $\lambda = 1.12$, doublet rate = 0.30, false negative rate = 0.30). The results for v-measure that we obtained are presented in the figure shown below. As can be observed, ddClone outperforms OncoNEM in this simulation setting and the obtained results are very similar to the results presented in the original ddClone publication. (average v-measure for ddClone is 0.7555 whereas average of this measure for OncoNEM is 0.7224).

*3. The authors used two “old” biological datasets (2014), and newer ones have become available. It might be good to run the method on newer datasets, such as the ones used in the SiFit paper.

We thank Luay for this comment. We are aware that we have been using older datasets. This choice was influenced by the quality of the bulk sequencing data, as we believe B-SCITE works best with high read depths. We have since added two more recent datasets from colorectal cancer patients with liver metastases (Leung et al. 2017). While these data are not ideal for B-SCITE (limited coverage in the bulk sequencing, aneuploidy), it provides data from two bulk samples per patient which allowed us to test our extension of B-SCITE towards multiple bulk samples on real data. We report the results of this new analysis in detail in the Results section (last subsection “Colorectal cancer”, page 7) and the newly added Figure 8.

*4 In most cases, the mutations in single-cell and bulk may differ. Sometimes, bulk data may have higher number of mutations being detected compared to targeted SCS data. Whether one should use the full set of bulk mutations or the set of mutations common to both is not clear. In particular, with respect to the proposed method/study, it is unclear what role (can this be quantified?) the SCS and bulk data play in the inference. That is, how much is the bulk data contributing to the inference, and how much is the SCS data contributing? Furthermore, the authors mention that both scores derived from SCS and bulk data are on the same scale, and that they weight them equally (the ω parameter). However, it is unclear why both scores are on the same scale. It would be good to report on the possible range of each score.

We thank Luay for picking up this point, and we have included the range of each score, as well as their ratio as a way of estimating the contribution from each data type. For the leukemia patients, this ratio turns out to be around 1. In general, of course, as Luay notes, this does not need to be the case since the scale depends on the coverage and number of cells. For the TNBC data, we observe a ratio of around 2800, for example.

With regards to whether to use the full set of mutations, or only those detected in single cells, we used the common mutations when comparing to other single-cell methods, and the full set when comparing to the bulk methods to make the results directly comparable.

*5. The authors mentioned the affect of the number of single cells to the performance, i.e., the larger the number, the higher the affect it would have on the inference. However, the authors didn't report the effect of the sequencing depth of the bulk tissue, which could be done by a simulation study. A plot of the accuracy with the sequencing depth of bulk tissue might be beneficial to the community in informing the extent to which to sequence a patient.

We thank Luay for raising this point. We have added a simulation study for 25, 50 and 100 single cells, three coverages, and 1, 2 and 4 bulk samples (Figures A25-A27). Interestingly we find better performance with more bulk samples at a lower coverage than a single sample at higher coverage, which can indeed be informative for designing sequencing strategies.

*6. The authors report the best tree, i.e., the tree with the highest likelihood. However, other trees might shed light on the quality/significance of the top tree. For example, it would be good to report the tree with the second highest likelihood and contrast it to the best.

We have extended our implementation so that the second best tree is now also reported in addition to the best one. Using this new implementation, we have compared the two best clonal trees reported for a subset of simulations (parameters same as in Figure A2). In order to perform direct comparison of the two inferred trees, we used MLTED-normalized similarity measure and MLTED-distance measure (for the definition of these two measures please see <http://drops.dagstuhl.de/opus/volltexte/2018/9324/>). Our results show that the average MLTED-normalized similarity measure between the second best and the best clonal trees reported by B-SCITE equals 0.997, while (and perhaps because of the fact that) MLTED-distance between the two trees equals 0 in 95% cases. This implies that in the vast majority of the cases the two most likely trees have the same basic topology - with slight differences with respect to whether certain mutations are clustered in a single vertex or are spread over two or more vertices in a non-branching path.

*7. The authors conducted all simulations under the infinite sites assumption. While their model/method employ this assumption, could they simulate data that violate the assumption and study how the method performs on such data? This is important especially in light of the recent study by Beerenwinkel and colleagues demonstrating violation of this assumption in empirical datasets.

We thank Luay for raising this important point. We have investigated this issue in a new simulation study and observed that adding violations of the infinite sites assumptions to the data has only a minor effect on the accuracy of inferring the unaffected mutations (Figure A24).

*8. Minor: First word on page 3: “complimentary” → “complementary”.

We thank Luay for pointing to this misspelling. It is now fixed.

Response to Reviewer 2:

We thank Reviewer 2 for their positive feedback and suggestions to improve our work.

*1. Extend B-SCITE to support multiple bulk samples from the same tumor.

As mentioned by the authors a single bulk sample does not allow one to identify branching evolution. However, with multiple bulk samples such patterns start to become apparent. I encourage the authors to extend their model and algorithms to support multiple bulk samples from the same tumor. The use of multiple spatial/temporal bulk samples is becoming increasingly common place.

We thank the reviewer for this valuable suggestion. We have now extended our model and implementation to account for cases where sequencing data from multiple bulk sample is available. As results show (Figures A25-A27), the inference improves with more cells, more samples and higher coverage, but interestingly we find better performance with more bulk samples at a lower coverage than a single sample at higher coverage. This could be informative for designing sequencing strategies.

*2. Improved/additional simulations.

Using simulated data, the authors compare B-SCITE against ddClone, OncoNEM and SCITE. To assess B-SCITE's improvement over the current state of the art, I would like to see a comparison against methods that only use bulk sequencing samples (a selection from CTPsingle, CITUP, AncesTree, PhyloWGS, etc.).

We have now added comparisons to PhyloWGS, a method which can exploit sequencing data from multiple bulk samples. In the publication introducing ddClone (Salehi et. al, Genome Biology, 2017) it has already been shown that ddClone outperforms PhyloWGS in terms of v-measure and, as we have seen that B-SCITE consistently outperforms ddClone in terms of clustering accuracy, we focused here on the comparison of the inferred phylogenetic trees. Results of these comparisons are

presented in Figures A25-A27. In summary, we find that both B-SCITE and PhyloWGS benefit from the use of multiple samples. While B-SCITE generally outperforms PhyloWGS, it is possible to construct settings where the bulk-only method is favoured, namely, as expected, a combination of multiple bulk-samples sequenced at high-coverage with a small number of single-cell data that badly reflect the tumour composition (small λ).

*3. In addition, I would like to see simulations that assess the interplay between the number of bulk and single-cell samples. Moreover, the current simulations seem to assume a coverage of 10,000X. It would be good to assess B-SCITE's robustness in the case of lower coverage (e.g. WGS: 60X) and varying SCS false positive and false negative rates.

We investigated these dependencies in our new simulation study where we look at 25, 50 and 100 single cells, three coverages, and 1, 2 and 4 bulk samples (Figures A25-A27). At lower coverage, we observe that the accuracy decreases, but this can be more than compensated for with multiple bulk samples. False negatives play a fairly weak role in B-SCITE's robustness, but we do see some decrease with elevated false positive rates (Figures A8 and A9 along with A16 and A17).

*4. In addition to the V-measure, it would be a good idea to measure to clustering accuracy using more direct measures such as the Rand index.

We included the adjusted Rand index for several of the simulations (Figures A5-A7) for which we had used the V-measure (Figures A2-A4), with results that are very similar. We chose the adjusted Rand index over the unadjusted version, since the adjustment makes the standard values of 0 and 1 interpretable.

*5. The effect of copy-number aberrations seems to be only measured in terms of co-clustering accuracy. What happens with the other measures? Also, how is the difference defined (y-axis of Fig. A11)?

We see a similar pattern with the other distance measures and have included the corresponding plots (A18-A23). For each simulation, we have generated data from the same tree with and without CNAs. The difference depicted in the plots is relative to the accuracy measure with no CNAs.

*6. Finally, it is not completely clear how CNAs were simulated. (e_i, f_i) seem to denote changes in the variant and reference allele counts. What are the initial values? Does this number ever become negative?

We thank the reviewer for pointing out this lack of clarity and hope that we have now clarified the previous ambiguities by rewriting the first part of the description in the Appendix section "Simulation of CNA events". We have now added that prior to a CNA event each cell has either two reference or one reference and one variant copy of region R (depending on whether the cell harbors mutation M). If a CNA event occurs in a cell having both reference copies of R, then e must be 0 (i.e. no gain or loss of variant copy is possible since such copy does not exist in the cell) implying that the number of variant copies of R in cells affected by the CNA event is $0+e=0$, whereas the number of reference copies is $2+f$. Similarly, if an event first occurs in a cell having one reference and one variant copy of R, then the total number of variant and reference copies of this region in cells affected by the CNA event is $1+e$ and $1+f$, respectively. Note that, due to the assumption $|e|+|f|=1$, all of the numbers $2+f$, $1+e$ and $1+f$ are non-negative. Also note that e and f can take negative value of -1 which represents deletion of the corresponding copy of region R (the only exception when one of these two numbers can not be negative is the case when e must be 0 as discussed above).

Minor comments of Reviewer 2

* Last sentence of abstract: "... more realistic tumor histories." Please be more specific. We rephrased this sentence. It now states the inferred tumour phylogenies are in high concordance with expert generated tree (where available).

* Please be consistent in the notation of matrix entries ($A_T[i,j]$ vs $D_{\{i,j\}}$)
We have now updated our notation so that $D[i,j]$ is used instead of $D_{\{ij\}}$.

* Typo/awkward phrasing: "When $L_T(v_i)$ denotes the set of mutations..."
We have now revised (and also reordered some paragraphs) in the section "Tree models of tumour evolution" and this phrasing was removed.

* Typo on page 7: subclons
We thank reviewer for pointing to this misspelling which is not present in the updated version of the manuscript.

* For completeness, it would be good to give a derivation of (5) in the supplement.

We have added section A3 (in the Appendix) where more details of the derivation of (5) are provided.

Response to Reviewer 3:

We thank the reviewer for their positive feedback and suggestions to improve our work.

*1. The authors make a clear case for the method, but the study does not go beyond methodological advances. The limited amount of analysis is confined to two patients, reflecting the difficulty to obtain, and therefore the rarity of, applicable data. I would be much more enthusiastic about the paper if novel data had been generated or if much more data had been analyzed, and application of the method had generated some insight.

We thank the reviewer for this comment. We have now extended our analysis to five patients (2 leukemia, 1 breast cancer, 2 colon cancer). Both labs involved in this work are purely computational, but we would be happy to collaborate with an experimental lab in the future to generate novel data. In the discussion of the inferred mutation histories, we have mentioned potential biological insights such as the clear separation of a primary and metastasis branch in one of the colorectal cancer histories. In general, the focus of this work is the methodological advance, which facilitates improved tumor phylogeny reconstruction from joint bulk and single-cell data. In addition, our work enables, for the first time, the design of optimal tumor sampling and sequencing strategies across multiple bulk samples and single cells.

*2 One major concern that remains in any method that heavily relies on VAF to infer somatic lineage trees is the confounding influence of copy number variation on VAF. This is an issue that has not been solved and B-SCITE is as susceptible to it as any other method that relies on VAF. For example, VAF of close to 100% (e.g., MAL2 and RYR3 in Fig 5a) is almost certainly due to loss of the wild type chromosome (LOH); more insidiously, a VAF of 1/3 or 2/3 is almost certainly due to either the wild type or the mutant chromosome having duplicated, respectively. Unless these events are explicitly modeled, VAF-based inference cannot be considered reliable because it misinterprets genomic events (i.e., copy number changes) as differences in cellular representation, and thus gives predictably erroneous results. Given the prevalence of copy number

changes in cancer, especially in solid tumors, this is a major issue for any such methods, including B-SCITE.

We thank the reviewer for pointing out the confounding influence of copy number alterations. This is a well-known issue affecting the accuracy of VAF based clonal inference. It is typically obviated by restricting the inference to mutations in presumably copy number-neutral regions. This process is of course not perfect, and we can still expect a number of mutations from copy number altered regions in our datasets. While B-SCITE does not specifically solve this issue, our extensive simulation studies show a remarkable robustness of our inferences to realistic rates of copy number alterations (see Figures A18 through A23). The reason for this robustness is in the additional use of single-cell data which is not VAF based. As can be seen in the depicted trees, B-SCITE does not strictly sort mutations by decreasing frequency. Such a deviation is bound to happen whenever a strong signal from the single-cell data suggests an ordering that differs from strictly decreasing VAFs. The deviating VAFs can be either due to technical issues (such as low coverage at the position) or indicate instances of copy number alterations.

We also believe the previous version of our manuscript lacked clarity in pointing out the difference between (i) variant allele frequency (VAF) and (ii) cellular frequency/prevalence. The former is defined as $VAF = v/(v+r)$, where v denotes the number of variant reads and r denotes the number of reference reads). The latter is the expected fraction of cells harboring mutations in a copy number neutral region (i.e. $2 \cdot VAF$). We changed the manuscript by using the term VAF over cellular frequency/prevalence wherever possible and changed the depicted percentages in the plots accordingly. We hope these changes address the reviewer's comment about a possible LOH event at MAL2 and RYR3. The VAFs of these mutations are around 48%, not close to 100%, thus suggesting them to be clonal heterozygous mutations in a diploid region.

REVIEWERS' COMMENTS:

Reviewer #1 (Remarks to the Author):

I very much appreciate the authors addressing my comments comprehensively through added simulations and results. I am very satisfied with the new version.

Best regards,
Luay

Reviewer #2 (Remarks to the Author):

The revised manuscript satisfactorily addresses my previous comments. This work is an important advance in the field, being the first method to jointly consider SCS and bulk data for tumor phylogeny inference, and will be of interest to the readership of this journal.

Mohammed El-Kebir